# Intervention Programs for First-Episode Psychosis: A Scoping Review Protocol

Marta Gouveia [1,2,3,*], Tiago Costa [3,4,5], Tânia Morgado [3,6,7,8], Francisco Sampaio [3,5], Amorim Rosa [7,9] and Carlos Sequeira [3,5,10]

1. Hospital Center of Tondela-Viseu, 3504-509 Viseu, Portugal
2. Abel Salazar Biomedical Sciences Institute, University of Porto, 4050-313 Porto, Portugal
3. CINTESIS@RISE, Nursing School of Porto (ESEP), 4200-450 Porto, Portugal
4. Hospital Center of Vila Nova de Gaia/Espinho, 4434-502 Vila Nova de Gaia, Portugal
5. Nursing School of Porto, 4200-072 Porto, Portugal
6. Pediatric Hospital of the Centro Hospitalar e Universitário de Coimbra, 3000-602 Coimbra, Portugal
7. Health Sciences Research Unit—Nursing (UICISA: E), Nursing School of Coimbra, 3000-232 Coimbra, Portugal
8. School of Health Sciences, Polytechnic of Leiria, Campus 2, Morro do Lena, Alto do Vieiro, Apartado 4137, 2411-901 Leiria, Portugal
9. Nursing School of Coimbra, 3000-232 Coimbra, Portugal
10. Research Unit of the Nursing School of Porto, 4200-072 Porto, Portugal
* Correspondence: martagou@hotmail.com; Tel.: +351-969654461

**Abstract:** Early intervention programs for first-episode psychosis aim to intervene in the early stages of the disease. They are essential to prevent and delay the progression of the illness to a more advanced stage, but information about their characteristics is not systematized. The scoping review considered all studies of first-episode psychosis intervention programs, regardless of their context (hospital or community), and explored their characteristics. The scoping review was developed according to the Joanna Briggs Institute methodology and PRISMA-ScR guidelines. The PCC mnemonic (population, concept, and context) addressed research questions, the inclusion and exclusion criteria, and the search strategy. The scoping review sought to identify the literature that meets the predefined inclusion criteria. The research was conducted in the following databases: Web of Science Core Collection, MEDLINE, CINAHL Complete and PsycINFO, Scopus, Cochrane Library, and JBI Evidence Synthesis. The search for unpublished studies included OpenGrey (a European repository) and MedNar. It used sources in English, Portuguese, Spanish, and French. It included quantitative, qualitative, and multi-method/mixed methods studies. It also considered gray or unpublished literature. After removing duplicates, two independent reviewers extracted the relevant information after selecting the articles. If there were disagreements, a third reviewer was used. The researchers have developed a tool based on the JBI model that will allow them to extract the relevant information for the review. The results are presented schematically in narratives and tables. This scoping review maps first-episode psychosis intervention programs by identifying their characteristics, participants, and specific implementation contexts and allows researchers to create multicomponent programs tailored to different contexts.

**Keywords:** psychosis; early intervention; education; evidence-based practice

## 1. Introduction

Psychosis is a mental illness characterized by altered thinking and perception [1]. It is a broad concept that describes a condition where different symptoms, such as hallucinations, delusions, and disorganized thinking, may be present. It may be associated with changes in mood and cognitive functioning [2–4] (such as memory functioning, frequent changes in concentration, disorganized ideas and behavior, social isolation, or avolition) [3]. The age range where it appears is variable but usually appears in the adolescent-young adult age [1].

The study of psychotic illness has led to understanding its nosological presentation along a continuum. The staging of psychotic illness allowed the different phases of the disorder to be distinguished and interventions to be adapted to each one. Thus, a preventive approach was possible [5], where the possibility of delaying the transition to more advanced stages of the disease or even the return to the previous state was recognized, thus improving the initial prognosis, often unfavorable, which would be associated with this type of disorder [2,5–7].

The staging concepts of this mental illness differ from establishing a conventional diagnosis. They allow us to define the extent of the progression of the disease and where a person lies along the continuum of the course of a mental illness [6,8]. Mapping the development, progression, and extent of mental illness over time allows for selecting more appropriate treatment and intervention in the early stages of illness, resulting in greater efficacy than later on. The result is an adequate treatment for each stage to prevent and delay progression to more advanced stages [9].

Early intervention in psychosis is based on two central constructs—the critical period hypothesis, developed by Birchwood and colleagues, [10] and issues associated with the duration of untreated psychosis (DUP). The critical period hypothesis proposes that deterioration occurs aggressively during the early years of psychosis, with relative subsequent stability, and treatment is much more effective in the early stages of illness (from 3–5 years). The intervention developed at this stage interferes with the disease's prognosis and evolution, with better results than in the later phases. The DUP (period from the onset of symptoms to the start of treatment) assumes that the person's potential for recovery decreases as the DUP increases. DUP is an extremely important variable because it has become a predictor of intervention outcomes. Thus, interventions that reduce the DUP and interrupt early deterioration may have long-lasting beneficial effects [11–13].

Following on from this, another important concept is that of early intervention (EI). By early intervention, we mean differentiated and appropriate intervention for each stage of the disease. It aims to minimize the associated negative effects and improve prognosis, either through recovery or by delaying the transition to more advanced stages of the disease [14]. EI is a key element of recovery, being more efficient than standard care services [15]. Considering this disease's course and staging concepts, from the prodromal phase until rehabilitation, intervention in psychotic disorders should occur as early as possible [6].

The first psychotic episode is understood to be the inaugural period where the onset of symptomatology lasts for at least one week, disrupting the person's functioning [3,4]. As we said above, it corresponds to the early onset of symptoms, followed by a critical period of two to five years, which is crucial for recovery and prognosis [1,2,7]. Thus, interventions in the critical phase are essential. During the first psychotic episode, and especially throughout the critical period, the family plays an important supportive role. Yet, there is a major impact on their lives; there is great distress for family carers, especially those who live with the person, as they will not have had previous experience caring for a person with psychosis [16]. In addition to this important factor, the caregivers face a challenging period where, in addition to having little knowledge of the mental health situation, they are faced with a sense of grief or loss. This factor is very important as it interferes with their expressed emotion towards the patient. The literature has described the importance of the family environment in the recovery process. Family interventions for people experiencing a first psychotic break contribute to improving their expressed emotions and psychological stress [17]. According to Petrakis and Laxon (2017), psychoeducational interventions for caregivers are a predictor for better relapse prevention rates as well as the promotion of their recovery. On the other hand, it is a strategy to prevent isolation and provide relevant support [18].

In recent years, teams dedicated to specific interventions in first-episode psychosis have proliferated, showing more promising results when compared to those obtained in standard services [1,15,19]. These teams promote recovery and a productive life. They offer

appropriate interventions aimed at ameliorating clinical outcomes, such as the impact of symptoms, improving functioning, and reducing relapse with a view to recovery as early as possible [19–21]. Recovery encompasses the issues of clinical recovery and personal recovery. Clinical recovery is the process by which a person recovers from previous health [22,23]. It is associated with recovery at the level of psychotic symptomatology and at the level of functioning (a type of recovery where the person can develop practical life skills, responding to and fulfilling social and occupational roles [2]. Personal recovery is associated with the process by which the person progresses through illness and adapts to their new state. The symptoms of the disease may or may not remain, but the person feels that his/her life has meaning and is capable of responding to his/her life project adapted to his/her circumstances, regardless of the disease [23,24].

Early intervention teams are multidisciplinary teams with well-defined goals of early detection to reduce the period of untreated psychosis (due to the risk of a worse prognosis) [12]. They work towards a global concept of recovery (functional and personal recovery) and relapse prevention and aim to minimize the DUP's effects by reducing the illness's severity and impact. Using a comprehensive approach to the situation, they provide a whole set of structured care with community involvement (pharmacological, psychological, and social interventions) to individuals and caregivers [1–3,15,25]. Early intervention programs for psychosis should develop interventions with the caregiver and should meet their needs. They should advocate close contact with them to provide them with the necessary strategies to meet their needs, such as psychoeducational interventions or family counseling [2,4]. Besides these, behavioral family therapy (BFT) and systemic family therapy [4] are also mentioned.

Recent literature has shown high recovery rates, which leads to a focus on early intervention programs with this type of intervention [26,27]. These teams are an alternative rather than an addition to standard psychiatric care [1].

Early intervention in psychosis in the form of early intervention teams is widespread and adopted in several countries [1,28]. Addington and colleagues sought to identify the essential components of the services of early intervention in psychosis. These authors highlighted the evidence-based practices of these services, which are in line with those previously issued by the World Health Organization and the International Early Psychosis Association [25]. They stressed the importance of the intervention models of the services (psychiatrist as part of the team, duration of treatment, team training, weekly team meetings, crisis intervention, and dissemination of results) [29]. The objectives of early intervention teams in first-episode psychosis are: (a) to intervene in the early stages of illness, reducing the DUP as much as possible; (b) to offer the person a complete treatment (integrated and comprehensive model) that includes medication, psychological therapies, and psychoeducation for the family; (c) to promote recovery by improving the prognosis and course of the illness. In this sense, evidence shows that early intervention in psychosis is associated with (a) early detection and reduced DUP; (b) less severe symptoms and fewer suicides and deaths; (c) lower risk of progression to more advanced levels of psychosis; (d) higher rates of recovery and relapse prevention; (e) fewer relapses and hospitalizations; (f) higher patient satisfaction and involvement with services; (g) better levels of overall functioning and quality of life; (h) lower costs [2–4,26,30,31]. The strategy inherent to early intervention is patient-centered and comprehensive. It is based on a multidisciplinary assessment, identifying needs that leverage a psychosocial approach, [2,3] treatment adherence, and psychopharmacological, family, and vocational interventions [25]. Early intervention teams should offer appropriate interventions to improve the impact of symptoms, improve functioning, and reduce relapses with a view to recovery as early as possible. They should provide a broad range of interventions to [7,32] minimize the effects of the DUP by reducing the severity and impact of the illness [21].

The interest in studying psychoses has grown over the past decades [33]. De Maio and colleagues, in their review, refer to the existence of several programs developed in different countries which have obtained good results in their interventions, namely in terms of the

reduction in psychotic symptoms, overall functioning, quality of life, and reduction of hospitalizations [34]. Early intervention services emerged in Melbourne, Australia. It has been followed by the United Kingdom, America, and Asia [20].

For the development of early intervention programs, guidelines can rely upon that attempt to list a whole set of recommendations on the key components for their success. Examples include the International Early Psychosis Association Writing Group, [35] Orygen, [2] HSE (Health Service Executive) [4], and NASMHPD (National Association of State Mental Health Program Directors) [30]. The literature has described as components of intervention programs, comprehensive psychotherapeutic interventions (individual, group, and family) as fundamental in the process of patient recovery and promotion of quality of life [14], and several reviews seek to list the essential components of early intervention programs [29,32,34,36]. Although these documents attempt to describe the interventions developed, a challenge remains in their applicability. Thus, although evidence-based interventions are already described, early intervention services present different models of service delivery [34]. In her systematic review concerning intervention model delivery, Behan (2016) says that early intervention models are heterogeneous. He states that these are aligned in a continuum and describes three, namely, the specialist or stand-alone, the enhanced community health care, and the hub and spoke. The implementation of early intervention programs is challenging and requires adaptations according to the contexts where the program is implemented. Thus, these three models differ in the area in which they operate. For example, in the hub and spoke model, health professionals are responsible for early intervention in psychosis in a well-defined geodemographic area, usually urban, close to central mental health care centers. In the hub and spoke model, in an attempt to access rural areas, the team is divided into a central service—hub—and several spokes that seek to intervene locally in more remote areas. The enhanced community mental health care model refers to a model where the professionals of the community mental health teams add early intervention in psychosis in addition to their roles in their teams. From these models, it can be inferred that the stand-alone model will be the most effective one, but it presents difficulties in terms of its accessibility to more remote areas. Although the different models differ in their operationalization and adapt to local demands, [36] the clinical results point to the effectiveness of the intervention of these specialized services compared to common mental health services [1,15].

Still, we realize the variability of the goals and form of implementation of the programs, [34] coupled with the lack of psychometric data, makes it difficult to describe pathways to care. Currently, some reviews attempt to describe pathways to care. However, the lack of psychometric data and the variables inherent to the different programs make it difficult to describe them [37]. Other reviews refer us to the characteristics of the model of care inherent in early intervention services to develop fidelity scales to standardize its implementation [38,39]. In this sense, the concerns surrounding early intervention programs are based not only on their proliferation but also on their quality indices [20,40]. While the effective outcomes of early intervention programs are understood and their proliferation is noted, they remain an exception rather than a component of mainstream mental health services, [20] with a wide dispersion in accessibility to these services, as reported by Petrovic and colleagues in their study for Europe, [41] Lilford about low and middle-income countries [40], and Aceituno in America Latina [42]. On the other hand, we find that in high-income countries, program development has been extensive [43], while in the middle to low-income countries, progress is very slow [30,40]. Early intervention services are well established in Australia, Canada, New Zealand, the USA, Singapore, Hong Kong, and Japan. In Europe, the United Kingdom, Denmark, and Switzerland stand out, [20] but their development is uneven in different countries and even in those with early intervention services, there are inequalities in access to care, with the development of geographically dispersed services [41].

There remain inherent difficulties in implementing these services regardless of the understanding that the benefits associated with early intervention in psychoses are consis-

tent and more effective than standard care [1,15]. O'Connell and colleagues state that the implementation of early intervention services is inconsistent and fragmented and point to issues in health systems and services or linked to the services themselves as barriers and facilitators to their implementation [44].

The literature has tried to understand what influences the DUP, and we realize that the concept does not refer only to the act of help-seeking. In addition to being culturally determined [40], access to often underfunded care for LMICs significantly interferes with greater DUP [45]. In this regard, while intervention models are important, there is a need for continued funding to make them viable [46], given that delays in treatment initiation are associated with worse treatment response and increased disability for people in LMICs [43]. Funding for mental health (and also for early intervention services) are below needs in these countries, [46] implying a wide dispersion in the accessibility of care, as reported by Petrovic and colleagues [41] and Lilford [40].

In LMIC, there are basic issues that point to the insufficiency of basic mental health care. However, intervention in psychosis is inadequate, not only due to lack of services but also to low funding and shortage of health professionals; Singh and Javed (2020) state that the basic principles of early intervention adopted in countries with more resources should be incorporated into the different levels of care, wherever they exist [47].

Given the importance of the topic under study, we found the need to adapt early intervention services to the different realities (social and political factors and health services structures) that promote and facilitate its applicability in all settings.

This review believes that there is a gap in the way programs describe their service configuration characteristics; thus, there is a need to clarify how these programs are structured (objective of the intervention; frequency; type of intervention; intervention facilitators; assessment; implementation context), thus avoiding knowledge dispersion (description of the interventions in different documents) and allowing for the appropriate selection of interventions in the structuring of programs that meet the constraints of each reality.

A preliminary search was performed in the Cochrane Database of Systematic Reviews, PROSPERO, MEDLINE, and JBI Evidence Synthesis. Although there is literature on the subject, no scoping review or systematic review responds to the specificities of the topic under study. Thus, it is vital to map the characteristics of the programs to facilitate their development and dissemination. Effective mapping of the different intervention programs will be crucial to explore the different programs, given the importance of early intervention in first-episode psychosis. Thus, the scoping review's objective was to map the programs considering their characteristics and all implementation contexts, whether hospital or community.

*Aim*

This scoping review aimed to map first-episode psychosis intervention programs by identifying their characteristics, participants, and specific implementation contexts.

## 2. Materials and Methods

This review was conducted according to the JBI scoping review methodology [48]. The Preferred Reporting Items for Systematic reviews and Meta-Analyses extension for Scoping Reviews (PRISMA-ScR) checklist was used as a matrix for structuring the article under study [49]. Considering the mnemonics Population (P), Concept (C), and Context (C), the inclusion and exclusion criteria for the structuring of the research strategy were analyzed. The review considered literature that refers to intervention (C) programs (C) applied to individuals with the first psychotic episode and family members (P) without the restriction of contexts, such as urban, rural, hospital settings, and community teams. The review protocol was registered in the Open Science Framework on February 26, 2022 [50]. OSF Registration Doi: 10.17605/OSF.IO/ZY9QM.

*2.1. Review Questions*

i.    Which early intervention programs are implemented in service users and family members with first-episode psychosis?

ii.   What are the characteristics of the intervention programs? The characteristics of the intervention programs are: name of the program; the objective of the intervention; frequency; type of intervention; intervention facilitators; evaluation; implementation context.

iii.  In what settings (outpatient, inpatient, community) are the programs implemented and evaluated?

iv.   Who is the target (patient, family, family, and patient) of the intervention programs?

*2.2. Inclusion Criteria*

2.2.1. Population

The review included individuals with symptoms associated with first-episode psychosis. The following designations were used: "first-episode psychosis", "recent onset psychosis", "early onset psychosis", and "early psychosis".

Patients in the early stage of psychosis were included, and individuals diagnosed with organic psychosis were excluded.

The review included caregivers. Caregivers are the first- or second-degree relatives who provide care, contact, or live with the person with first-episode psychosis.

2.2.2. Concept

Intervention programs are programs with a specific design for intervention in first-episode psychosis and early onset psychosis—with various types of intervention (psychotherapeutic, psychosocial, vocational, cognitive-behavioral, and psychoeducational). Their work focuses on providing care to the person with first-episode psychosis and the family caregiver. Interventions carried out in a generalized manner in non-specific consultations for the early phase of psychosis were excluded.

2.2.3. Context

This scoping review considered studies in all types of contexts. Hospital and community settings were included. The inclusion criterion was applied considering the objectives of the review.

2.2.4. Types of Sources

The scoping review considered quantitative, qualitative, and multi-method/mixed methods studies. Quantitative studies included experimental studies (randomized controlled trials, non-randomized controlled trials, and quasi-experimental studies) and observational studies (with descriptive, exploratory, and analytical designs). All systematic reviews were included, independently of the types of methods of search used. Further, also included was grey or unpublished literature, such as dissertations and theses, reports, government publications, documents from organizations, and guidelines. The sources of information in English, Portuguese, Spanish, and French were included, according to the level of language proficiency of the authors, without geographical or cultural limitations.

*2.3. Search Strategy*

The search strategy aimed to identify published and unpublished studies. An initial search of MEDLINE (PubMed) and CINAHL (EBSCO) was performed to locate articles on the topic under study. The titles and abstracts' analysis, according to the inclusion criteria as well as the index terms, allowed a complete search strategy on MEDLINE with full text (access via PubMed) (see Appendix A). The search strategy, including all identified keywords and indexing terms, was adapted for each selected information source, and article reference lists were included for additional articles.

*2.4. Information Sources*

Databases to be searched included Web of Science Core Collection (ISI Web of Knowledge), MEDLINE with Full Text, CINAHL Complete and PsycINFO (access via EBSCOhost Web), Scopus, Cochrane Library, and JBI Evidence Synthesis. The search for unpublished studies included OpenGrey (a European repository) and MedNar.

*2.5. Study Selection*

Articles were loaded into EndNote vX9 (Clarivate Analytics, Philadelphia, PA, USA), and duplicate studies were removed. According to the inclusion criteria, two independent reviewers proceeded with the study selection by reading the titles and abstracts, and after that, the articles were analyzed in full. Articles that do not have an abstract available were also included. Subsequently, two independent reviewers exhaustively assessed the full text of the selected citations that meet the inclusion criteria. If consensus was not reached in any of the steps, discussion or use of a third reviewer was used to reach consensus. Citations of eligible studies retrieved in full were imported into the JBI System for Unified Management, Evaluation, and Review of Information (JBI SUMARI) (Joanna Briggs Institute, Adelaide, Australia). Excluded full-text articles were registered and made known in the scoping review. The research results are fully described in the final review and presented in a PRISMA-ScR flowchart [50].

*2.6. Data Extraction*

Data were extracted from articles included in the scoping review by two independent reviewers using a data extraction tool developed by the reviewers based on the JBI instrument for extracting details of the studies, characteristics, and results [50]. The extracted data include specific details about the intervention programs. The data extraction tool was developed (see Appendix B). The draft data extraction tool was modified and revised as needed throughout the data extraction process, taking into account each item included. All changes are described in the scoping review. Disagreements between reviewers were resolved through discussion or a third reviewer. Whenever necessary, if there are missing data in articles, the authors will be contacted to request them.

*2.7. Data Presentation*

Data are presented in visual representations, narratives, and tables. Data extracted from the different studies include the title, author, year of publication, country of origin, type of study, and objectives. The data extracted regarding the studies include participant characterization, program characteristics, and implementation context.

Participant characterization indicates the diagnosis, age, and intervention target (patient, family, family, and patient). Regarding program characteristics, the name of the program, the objective of the intervention, the frequency (indicating the number, duration, and periodicity of sessions, and the follow-up period in the program), the type of intervention (intervention strategy and content), the intervention facilitators, and evaluation are presented. Data presented also include the implementation context, specifically the geographical area and description of the location and environment where it takes place, whether hospital, community, or outpatient. This topic mentions the number of participants (group or individual intervention).

## 3. Expected Results, Conclusions, and Implications

The study of early intervention has proliferated over the past 30 years. It has been found that early intervention teams in psychosis have evolved and contributed to the burden of disease that can translate into gains for people's health.

Recent literature has shown that early intervention in psychosis has promising results in the person's recovery. Although it is perceived that early intervention programs have promising results in the recovery process, access to this type of care is still dispersed. On the other hand, there are several programs with different objectives and variations in their

implementation that emphasize different interventions. In this sense, it is important to map the programs so that their applicability to different contexts can be adapted and facilitated. There is dispersion in the accessibility to this type of care since it is observed that it is not a standard component of mental health services, besides the need to adapt programs to different realities and environments. With this review, we hope to map intervention programs for first-episode psychosis and its structuring characteristics.

It will allow for advances in clinical practice and research because it will allow researchers and health professionals to design a new multicomponent program.

On the other hand, given the different contexts (political, social, and cultural), it may have beneficial contributions at the management level due to the possibility of selecting appropriate interventions for each reality and subsequent implementation.

**Author Contributions:** Conceptualization, M.G., T.C., F.S., A.R. and C.S.; methodology, M.G., T.C., T.M., F.S., A.R. and C.S.; validation, M.G., T.C., T.M., F.S., A.R. and C.S.; writing—original draft preparation, M.G., T.C., F.S., A.R. and C.S.; writing—review and editing, M.G., T.C., T.M., F.S., A.R. and C.S. All authors have read and agreed to the published version of the manuscript.

**Funding:** This research received no external funding.

**Institutional Review Board Statement:** Not applicable.

**Informed Consent Statement:** Not applicable.

**Data Availability Statement:** Not applicable.

**Acknowledgments:** The authors would like to thank all the support and guidance from the Health Sciences Research Unit: Nursing (UICISA: E).

**Conflicts of Interest:** The authors declare no conflict of interest.

**Appendix A**

**Table A1.** Draft of search strategy to Medline (PubMed) [1].

| Search No. | Query | Records Retrieved |
|---|---|---|
| #1 | Search: ("first episode psychosis"[Title/Abstract] OR "First-episode psychosis"[Title/Abstract] OR "first episode psychoses"[Title/Abstract] OR "First-episode psychoses"[Title/Abstract] OR "first episode of psychosis"[Title/Abstract] OR "First-episode of psychosis"[Title/Abstract] OR "first episode of psychoses"[Title/Abstract] OR "First-episode of psychoses"[Title/Abstract] OR "early onset psychosis"[Title/Abstract] OR "early onset psychoses"[Title/Abstract] OR "early psychosis"[Title/Abstract] OR "early psychoses"[Title/Abstract]) AND (("Psychotherapy, Group"[Mesh] OR "Psychosocial Intervention"[Mesh] OR "Behavioral Symptoms"[Mesh] OR "Cognitive Behavioral Therapy"[Mesh] OR "Counseling"[Mesh]) OR ("early intervention"[Title/Abstract] OR "Group Psychotherapy"[Title/Abstract] OR "Group therapy"[Title/Abstract] OR "Cognitive behaviour"[Title/Abstract] OR "Cognitive behaviours"[Title/Abstract] OR "cognitive behavioral"[Title/Abstract] OR "Biopsychological interventions"[Title/Abstract] OR "Biopsychological intervention"[Title/Abstract] OR "Psychosocial Interventions"[Title/Abstract] OR "Psychosocial Intervention"[Title/Abstract] OR "Behaviour Therapy"[Title/Abstract] OR "Behaviours Therapy"[Title/Abstract] OR "behavioural therapy"[Title/Abstract] OR "Cognitive Restructuring"[Title/Abstract])) | 1753 |

[1] Search date: 28 December 2022

## Appendix B

**Table A2.** Draft of data extraction matrix.

| | | | |
|---|---|---|---|
| General study information | Title<br>Author<br>Publication year<br>Country of origin<br>Type of study<br>Objectives | | |
| Participants characteristics | Diagnosis, age<br>Intervention target (patient, family, family, and patient) | | |
| Program characterization | Program name | | |
| | Intervention Objective | To reduce psychotic symptomatology, increase knowledge, reduce stigma, deal with stress, etc. | |
| | Frequency | Number of sessions.<br>Duration of sessions.<br>Number of frequency of sessions.<br>Follow-up period in the program. | |
| | Intervention type | Intervention strategy | Psychoeducational, cognitive-behavioral, vocational intervention, etc. |
| | | Content | Topics addressed in the intervention. |
| | Intervention facilitators | The number of technicians involved in each intervention, their training, and other relevant characteristics. | |
| | Evaluation data | Methodology and frequency of evaluation of the intervention outcomes; identification of the instruments used (with at least construct validity). | |
| Implementation context | Geographical area and zone description—rural, urban | Inpatient, community, and outpatient settings. | |
| | Individual or group intervention (number of participants) | | |

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
