# Peer review of "Intervention Programs for First-Episode Psychosis: A Scoping Review Protocol"

_nursrep, doi:10.3390/nursrep13010026_

Round 1

Reviewer 1 Report

This is a very interesting paper presenting a protocol for a Scoping Review On Intervention Programs for first-episode of psychosis. The protocol is well-written, and of interest for the journal. However, before considering it for publication I recommend several minor changes.

Abstract. 

1- The first line of the abstract is a kind of aims or objectives, prior to the introduction description. I consider it can be removed.

2- The methods of the Scoping Review should be described in detail in the abstract, because the paper is based on a protocol. The methods should be further explained. Inclusion and exclusion criteria?

3- What about expected results? Are the authors hypothesizing differences between community and hospital based programs?

Introduction

1- I recommend to expand the introduction in terms of key interventions in that populations. Cognition, Functionality, recovery. I recommend to add some concepts and definitions in the introduction section.

2- The main objective of the scoping review should be expanded in a separate section, at the end of the introduction section. I recommend a new section called "1.1. Aims".

Methods

1- In the methods section, the authors should expand the inclusion and exclusion criteria. Inclusion criteria should not be only restricted to criteria of the population of study. 

Expected Results and implications.

1- This section should include some hypotheses about the expected results. Is there any hypothesis regarding cognitive results in patients attending early intervention programs? Is there any positive result in terms of functionality?

A conclusions sections is needed. This can be about the protocol.

Author Response

Response to reviewers’ comments

Dear Reviewer, thank you so much for all the comments and suggestions. We believe that it contributed to the improvement of the manuscript. Sincerely

Response to reviewers’ comments

Dear Reviewer, thank you so much for all the comments and suggestions. We believe that it contributed to the improvement of the manuscript. Sincerely

ANSWER: We thank the reviewer for this suggestion.

ACTIONS TAKEN

1 - As suggested, it has been deleted.

2- The methods of the Scoping Review should be described in detail in the abstract because the paper is based on a protocol. The methods should be further explained. Inclusion and exclusion criteria?

ANSWER: We thank the reviewer for this suggestion.

ACTIONS TAKEN

2 - The methodology to be adopted was further developed, explaining the issue of PCC mnemonics, according to the inclusion and exclusion criteria. The databases to be used and the type of articles to be included were added.

The following sentences had been improved:

“The PCC mnemonic (population, concept, and context) will address research questions, the inclusion and exclusion criteria, and the search strategy. The scoping review will seek to identify literature that meets the predefined inclusion criteria. The research will be conducted in the following databases: Web of Science Core Collection, MEDLINE, CINAHL Complete and PsycINFO, Scopus, Cochrane Library, and JBI Evidence Synthesis. The search for unpublished studies will include OpenGrey (a European repository) and MedNar. (…) It will include quantitative, qualitative, and multi-method/mixed methods studies. It will also consider gray or unpublished literature. (…) If there are disagreements a third reviewer will be used.”

3- What about expected results? Are the authors hypothesizing differences between community and hospital based programs?

ANSWER: We thank the reviewer for this suggestion.

ACTIONS TAKEN

3- It ends with what is expected to be obtained from the scoping review - a mapping of intervention programs that will allow the selection of appropriate interventions for each context.

The following sentences had been improved:

“This scoping review will map first-episode psychosis intervention programs by identifying their characteristics, participants, and specific implementation contexts and will allow researchers to create multi-component programs tailored to different contexts.”

Introduction

1- I recommend expanding the introduction in terms of key interventions in that populations. Cognition, Functionality, recovery. I recommend to add some concepts and definitions in the introduction section.

ANSWER: We thank the reviewer for this comment, this has been improved

ACTIONS TAKEN

1 - Throughout the introduction concepts such as the "critical period hypothesis" were introduced to reinforce the issues associated with the staging of mental illness.

The issue of the duration of untreated psychosis was also reinforced which allowed us to move on to develop the concept of early intervention and early intervention teams. In this context, it was important to define what is meant by recovery and functionality.

The following sentences had been improved:

“ It’s a broad concept that describes a condition where different symptoms such as hallucinations, delusions, and disorganized thinking may be present. It may be associated with changes in mood and cognitive functioning[2-4] (such as memory functioning, frequent changes in concentration, disorganized ideas and behavior, social isolation, or avolition).[3] The age range where it appears is variable but usually appears in the adolescent-young adult age. [1]

The study of psychotic illness has led to understanding its nosological presentation along a continuum. The staging of psychotic illness allowed the different phases of the disorder to be distinguished and interventions to be adapted to each one. Thus, a preventive approach was possible[5], where the possibility of delaying the transition to more advanced stages of the disease or even the return to the previous state was recognized, thus improving the initial prognosis, often unfavorable, which would be associated with this type of disorder. [2,5-7](…)

Early intervention in psychosis is based on two central constructs - the critical period hypothesis, developed by Birchwood and colleagues,[10] and issues associated with the duration of untreated psychosis (DUP). The critical period hypothesis proposes that deterioration occurs aggressively during the early years of psychosis, with relative subsequent stability, and treatment is much more effective in the early stages of illness (from 3-5 years). The intervention developed at this stage interferes with the disease's prognosis and evolution, with better results than in the later phases. The DUP (period from the onset of symptoms to the start of treatment) assumes that the person's potential for recovery decreases as the DUP increases. DUP is an extremely important variable because it has become a predictor of intervention outcomes. Thus, interventions that reduce the DUP and interrupt early deterioration may have long-lasting beneficial effects. [11-13]

Following on from this, another important concept is that of early intervention (EI). By early intervention, we mean differentiated and appropriate intervention for each stage of the disease. It aims to minimize the associated negative effects and improve prognosis, either through recovery or by delaying the transition to more advanced stages of the disease.[14] EI is a key element of recovery, being more efficient than standard care services.[15] (…)

The first psychotic episode is understood to be the inaugural period where the onset of symptomatology lasts for at least one week, disrupting the person's functioning.[3-4] (…) They offer appropriate interventions aimed at ameliorating clinical outcomes such as the impact of symptoms, improving functioning, and reducing relapse with a view to recovery as early as possible.[16-18] Recovery encompasses the issues of clinical recovery and personal recovery. Clinical recovery is the process by which a person recovers from previous health.[19,20] It is associated with recovery at the level of psychotic symptomatology and at the level of functioning (a type of recovery where the person can develop practical life skills, responding to and fulfilling social and occupational roles.[2] Personal recovery is associated with the process by which the person progresses through illness and adapts to their new state. The symptoms of the disease may or may not remain, but the person feels that his/her life has meaning and is capable of responding to his/her life project adapted to his/her circumstances, regardless of the disease.[20-21] (…) and aim to minimize the DUP's effects by reducing the illness's severity and impact. (…)The strategy inherent to early intervention is patient-centered and comprehensive. It is based on a multidisciplinary assessment, identifying needs that leverage a psychosocial approach,[2-3] treatment adherence, and psychopharmacological, family, and vocational interventions.[23] Early intervention teams should offer appropriate interventions to improve the impact of symptoms, improve functioning, and reduce relapses with a view to recovery as early as possible. They should provide a broad range of interventions to[7,30] minimize the effects of the DUP by reducing the severity and impact of the illness.[18]

2- The main objective of the scoping review should be expanded in a separate section, at the end of the introduction section. I recommend a new section called "1.1. Aims".

ANSWER: We thank the reviewer for this suggestion.

ACTIONS TAKEN

2 - Added a topic with the aim.

The following sentences had been improved:

“This scoping review will aim to map first-episode psychosis intervention programs by identifying their characteristics, participants, and specific implementation contexts.”

Methods

1- In the methods section, the authors should expand the inclusion and exclusion criteria. Inclusion criteria should not be only restricted to criteria of the population of study. 

ANSWER: We thank the reviewer for this suggestion.

ACTIONS TAKEN

1 - The methodology to be used is that of the JBI. Given this assumption, the recommended format requires that the inclusion criteria be in line with the PCC. Inclusion criteria regarding population, concept, context, and source types are part of the inclusion criteria according to this methodology.

Expected Results and implications.

1- This section should include some hypotheses about the expected results. Is there any hypothesis regarding cognitive results in patients attending early intervention programs? Is there any positive result in terms of functionality?

A conclusions sections is needed. This can be about the protocol.

ANSWER: We thank the reviewer for this comment, this has been improved

ACTION TAKEN

1- Given the type of study and scoping review, the main objective will be mapping programs. It was not intended to study the relationship between variables or differences between groups. Even so, I reinforce the idea that we expect to find several programs with different approaches and objectives.

In this topic, we added the conclusions that meet the importance of the review as a promoter of the development and growth of early intervention in psychoses, through the mapping of programs that may be facilitators for the creation of other culturally, politically, and socially competent programs in different areas.

The following sentences had been improved:

“Expected Results, Conclusions, and Implications

The study of early intervention has proliferated over the past 30 years. It has been found that early intervention teams in psychosis have evolved and contributed to the burden of disease that can translate into gains for people's health.

(…) Although it is perceived that early intervention programs have promising results in the recovery process, access to this type of care is still dispersed. On the other hand, there are several programs with different objectives and variations in their implementation that emphasize different interventions. In this sense, it is important to map the programs so that their applicability to different contexts can be adapted and facilitated. (…) besides the need to adapt programs to different realities and environments. (…)

It will allow for advances in clinical practice and research because it will allow researchers and health professionals to design a new multicomponent program.

On the other hand, given the different contexts (political, social, and cultural), it may have beneficial contributions at the management level, due to the possibility of selecting appropriate interventions for each reality and subsequent implementation.”

The following sentences had been improved:

“Introduction

During the first psychotic episode, and especially throughout the critical period, the family plays an important supportive role. Yet, there is a major impact on their lives; there is great distress for family carers, especially those who live with the person as they will not have had previous experience caring for a person with psychosis.[16] In addition to this important factor, the caregivers face a challenging period where, in addition to having little knowledge of the mental health situation, they are faced with a sense of grief or loss. This factor is very important as it interferes with their expressed emotion towards the patient. The literature has described the importance of the family environment in the recovery process. Family interventions for people experiencing a first psychotic break contribute to improving their expressed emotions and psychological stress.[17] According to Petrakis and Laxon (2017), psychoeducational interventions for caregivers are a predictor for better relapse prevention rates as well as the promotion of their recovery. On the other hand, it is a strategy to prevent isolation and provide relevant support.[18](…)

Early intervention programs for psychosis should develop interventions with the caregiver and should meet their needs. They should advocate close contact with them to provide them with the necessary strategies to meet their needs, such as psychoeducational interventions or family counseling [2,4]. Besides these, behavioral family therapy (BFT) and systemic family therapy[4] are also mentioned. (…)

The interest in studying psychoses has grown over the past decades.[31] De Maio and colleagues, in their review, refer to the existence of several programs developed in different countries which have obtained good results in their interventions, namely in terms of the reduction of psychotic symptoms, overall functioning, quality of life, and reduction of hospitalizations.[32] Early intervention services emerged in Melbourne, Australia. It has been followed by the United Kingdom, America, and Asia.[20]

For the development of early intervention programs, guidelines can rely upon that attempt to list a whole set of recommendations on the key components for their success.  Examples include the International Early Psychosis Association Writing Group,[33] Orygen,[2] HSE (Health Service Executive),[4], and NASMHPD (National Association of State Mental Health Program Directors).[28] The literature has described as components of intervention programs, comprehensive psychotherapeutic interventions (individual, group, and family) as fundamental in the process of patient recovery and promotion of quality of life[14]  and several reviews seek to list the essential components of early intervention programs.[27,30,32,34] (…)In her systematic review concerning intervention model delivery, Behan (2016) says that early intervention models are heterogeneous. He states that these are aligned in a continuum and describes three, namely the specialist or stand-alone, the enhanced community health care, and the hub and spoke. The implementation of early intervention programs is challenging and requires adaptations according to the contexts where the program is implemented. Thus, these three models differ in the area in which they operate. For example, in the hub and spoke model, health professionals are responsible for early intervention in psychosis in a well-defined geodemographic area, usually urban, close to central mental health care centers. In the hub and spoke model, in an attempt to access rural areas, the team is divided into a central service - hub - and several spokes that seek to intervene locally in more remote areas. The Enhanced community mental health care model refers to a model where the professionals of the community mental health teams add early intervention in psychosis, in addition to their roles in their teams. From these models, it can be inferred that the stand-alone model will be the most effective one, but it presents difficulties in terms of its accessibility to more remote areas. Although the different models differ in their operationalization and adapt to local demands,[34] the clinical results point to the effectiveness of the intervention of these specialized services, compared to common mental health services.[1,15]

Still, we realize the variability of the goals and form of implementation of the programs,[32] coupled with the lack of psychometric data makes it difficult to describe pathways to care. (…) In this sense, the concerns surrounding early intervention programs are based not only on their proliferation but also on their quality indices.[17,40] While the effective outcomes of early intervention programs are understood and their proliferation is noted, they remain an exception rather than a component of mainstream mental health services,[17] (…) and Aceituno in  America Latina.[41] (…) Early intervention services are well established in Australia, Canada, New Zealand, the USA, Singapore, Hong Kong, and Japan. In Europe, the United Kingdom, Denmark, and Switzerland stand out,[17] but their development is uneven in different countries and, even in those with early intervention services, there are inequalities in access to care, with the development of geographically dispersed services.[39]

(…)The literature has tried to understand what influences the DUP, and we realize that the concept does not refer only to the act of help-seeking. In addition to being culturally determined[40], access to often underfunded care for LMICs significantly interferes with greater DUP.[44] In this regard, while intervention models are important, there is a need for continued funding to make them viable,[38] given that delays in treatment initiation are associated with worse treatment response and increased disability for people in LMICs.[42] Funding for mental health (and also for early intervention services) are below needs in these countries,[38] implying a wide dispersion in the accessibility of care, as reported by Petrovic and colleagues[39] and Lilford.[40]

In LMIC there are basic issues that point to the insufficiency of basic mental health care. And, although intervention in psychosis is inadequate, not only due to lack of services but also to low funding and shortage of health professionals, Singh and Javed (2020) state that the basic principles of early intervention adopted in countries with more resources should be incorporated into the different levels of care, wherever they exist.[45]”

Reviewer 2 Report

Thank you for the opportunity to review, “Intervention Programs for First-episode Psychosis: a Scoping 2 Review Protocol”. This scoping review will likely have important implications for the early intervention of first-episode psychosis. The protocol has many strengths. The protocol itself is quite thorough. The introduction is well-written and demonstrates the need for such a scoping review. The search is expansive across multiple databases, inclusive of 4 languages, and includes the gray literature. I do have some suggestions to strengthen the protocol.

1.      The authors will use the JBI scoping review methodology. Since this is a protocol article, greater detail of the JBI methodology is necessary.

2.      Review question # IV—should be “Who is the target…” rather than “What…”?

3.      The sentenced “The characteristics of the intervention programs are: name of the program; objective of the intervention; frequency; type of intervention; intervention facilitators; evaluation; implementation context” seems to be misplaced on the same line as review question IV. Rather, it seems to better fit with review question ii.

4.      The expected implications could be expanded upon.

Author Response

Response to reviewers’ comments

Dear Reviewer, thank you so much for all the comments and suggestions. We believe that it contributed to the improvement of the manuscript. Sincerely

Reviewer 2

1.      The authors will use the JBI scoping review methodology. Since this is a protocol article, greater detail of the JBI methodology is necessary.

ANSWER: We thank the reviewer for this comment, this has been improved

ACTIONS TAKEN

  1. In the methodology topic, the questions inherent to following the methodology recommended by the JBI were developed, responding to the suggestions given.

The following sentences had been improved:

“Abstract

The PCC mnemonic (population, concept, and context) will address research questions, the inclusion and exclusion criteria, and the search strategy. The scoping review will seek to identify literature that meets the predefined inclusion criteria. The research will be conducted in the following databases: Web of Science Core Collection, MEDLINE, CINAHL Complete and PsycINFO, Scopus, Cochrane Library, and JBI Evidence Synthesis. The search for unpublished studies will include OpenGrey (a European repository) and MedNar. (…)It will include quantitative, qualitative, and multi-method/mixed methods studies. It will also consider gray or unpublished literature. (…) If there are disagreements a third reviewer will be used.

“2. Materials and Methods

Considering the mnemonics Population (P), Concept (C), and Context (C), the inclusion and exclusion criteria for the structuring of the research strategy will be analyzed. The review will consider literature that refers to intervention (C) programs (C) applied to individuals with the first psychotic episode and family members (P) without the restriction of contexts, such as urban, rural, hospital settings, and community teams. “

2. Review question # IV—should be “Who is the target…” rather than “What…”?

ANSWER: We thank the reviewer for this suggestion.

ACTION TAKEN

  1. It was changed in the text.

3. The sentenced “The characteristics of the intervention programs are: name of the program; objective of the intervention; frequency; type of intervention; intervention facilitators; evaluation; implementation context” seems to be misplaced on the same line as review question IV. Rather, it seems to better fit with review question ii.

ANSWER: We thank the reviewer for this suggestion.

ACTION TAKEN

  1. It was changed in the text

4. The expected implications could be expanded upon.

ANSWER: We thank the reviewer for this comment, this has been improved

ACTIONS TAKEN

  1. In this topic, the potential contributions that the scoping review could bring were developed a little more, namely in terms of contributions to clinical practice and contributions to research and management

The following sentences had been improved:

“3. Expected Results, Conclusions, and Implications

The study of early intervention has proliferated over the past 30 years. It has been found that early intervention teams in psychosis have evolved and contributed to the burden of disease that can translate into gains for people's health.

(…) Although it is perceived that early intervention programs have promising results in the recovery process, access to this type of care is still dispersed. On the other hand, there are several programs with different objectives and variations in their implementation that emphasize different interventions. In this sense, it is important to map the programs so that their applicability to different contexts can be adapted and facilitated. (…) besides the need to adapt programs to different realities and environments. (…)

It will allow for advances in clinical practice and research because it will allow researchers and health professionals to design a new multicomponent program.

On the other hand, given the different contexts (political, social, and cultural), it may have beneficial contributions at the management level, due to the possibility of selecting appropriate interventions for each reality and subsequent implementation.

Important note: as per the journal's guidelines, the main text of the protocol should be around 4000 words at minimum. The text has been expanded to answer this question.

The following sentences had been improved:

“Introduction

During the first psychotic episode, and especially throughout the critical period, the family plays an important supportive role. Yet, there is a major impact on their lives; there is great distress for family carers, especially those who live with the person as they will not have had previous experience caring for a person with psychosis.[16] In addition to this important factor, the caregivers face a challenging period where, in addition to having little knowledge of the mental health situation, they are faced with a sense of grief or loss. This factor is very important as it interferes with their expressed emotion towards the patient. The literature has described the importance of the family environment in the recovery process. Family interventions for people experiencing a first psychotic break contribute to improving their expressed emotions and psychological stress.[17] According to Petrakis and Laxon (2017), psychoeducational interventions for caregivers are a predictor for better relapse prevention rates as well as the promotion of their recovery. On the other hand, it is a strategy to prevent isolation and provide relevant support.[18](…)

The interest in studying psychoses has grown over the past decades.[31] De Maio and colleagues, in their review, refer to the existence of several programs developed in different countries which have obtained good results in their interventions, namely in terms of the reduction of psychotic symptoms, overall functioning, quality of life, and reduction of hospitalizations.[32] Early intervention services emerged in Melbourne, Australia. It has been followed by the United Kingdom, America, and Asia.[20]

For the development of early intervention programs, guidelines can rely upon that attempt to list a whole set of recommendations on the key components for their success.  Examples include the International Early Psychosis Association Writing Group,[33] Orygen,[2] HSE (Health Service Executive),[4], and NASMHPD (National Association of State Mental Health Program Directors).[28] The literature has described as components of intervention programs, comprehensive psychotherapeutic interventions (individual, group, and family) as fundamental in the process of patient recovery and promotion of quality of life[14]  and several reviews seek to list the essential components of early intervention programs.[27,30,32,34] (…)In her systematic review concerning intervention model delivery, Behan (2016) says that early intervention models are heterogeneous. He states that these are aligned in a continuum and describes three, namely the specialist or stand-alone, the enhanced community health care, and the hub and spoke. The implementation of early intervention programs is challenging and requires adaptations according to the contexts where the program is implemented. Thus, these three models differ in the area in which they operate. For example, in the hub and spoke model, health professionals are responsible for early intervention in psychosis in a well-defined geodemographic area, usually urban, close to central mental health care centers. In the hub and spoke model, in an attempt to access rural areas, the team is divided into a central service - hub - and several spokes that seek to intervene locally in more remote areas. The Enhanced community mental health care model refers to a model where the professionals of the community mental health teams add early intervention in psychosis, in addition to their roles in their teams. From these models, it can be inferred that the stand-alone model will be the most effective one, but it presents difficulties in terms of its accessibility to more remote areas. Although the different models differ in their operationalization and adapt to local demands,[34] the clinical results point to the effectiveness of the intervention of these specialized services, compared to common mental health services.[1,15]

Still, we realize the variability of the goals and form of implementation of the programs,[32] coupled with the lack of psychometric data makes it difficult to describe pathways to care. (…) In this sense, the concerns surrounding early intervention programs are based not only on their proliferation but also on their quality indices.[17,40] While the effective outcomes of early intervention programs are understood and their proliferation is noted, they remain an exception rather than a component of mainstream mental health services,[17] (…) and Aceituno in  America Latina.[41] (…) Early intervention services are well established in Australia, Canada, New Zealand, the USA, Singapore, Hong Kong, and Japan. In Europe, the United Kingdom, Denmark, and Switzerland stand out,[17] but their development is uneven in different countries and, even in those with early intervention services, there are inequalities in access to care, with the development of geographically dispersed services.[39]

(…)The literature has tried to understand what influences the DUP, and we realize that the concept does not refer only to the act of help-seeking. In addition to being culturally determined[40], access to often underfunded care for LMICs significantly interferes with greater DUP.[44] In this regard, while intervention models are important, there is a need for continued funding to make them viable,[38] given that delays in treatment initiation are associated with worse treatment response and increased disability for people in LMICs.[42] Funding for mental health (and also for early intervention services) are below needs in these countries,[38] implying a wide dispersion in the accessibility of care, as reported by Petrovic and colleagues[39] and Lilford.[40]

In LMIC there are basic issues that point to the insufficiency of basic mental health care. And, although intervention in psychosis is inadequate, not only due to lack of services but also to low funding and shortage of health professionals, Singh and Javed (2020) state that the basic principles of early intervention adopted in countries with more resources should be incorporated into the different levels of care, wherever they exist.[45]”
